pattern recognition

vector autoregressive, generative models, predictive models, art movements

**Author for correspondence:**
Edoardo Lisi
e-mail: edoardo.lisi95@gmail.com

# Modelling and forecasting art movements with CGANs

Edoardo Lisi[1], Mohammad Malekzadeh[3],
Hamed Haddadi[2], F. Din-Houn Lau[1] and Seth Flaxman[1]

[1]Department of Mathematics, and [2]Dyson School of Design Engineering, Imperial College London, London, UK
[3]School of Electronic Engineering and Computer Science Queen Mary University of London, London, UK

EL, 0000-0003-1186-6486; FD-HL, 0000-0003-1065-828X

Conditional generative adversarial networks (CGANs) are a recent and popular method for generating samples from a probability distribution conditioned on latent information. The latent information often comes in the form of a discrete label from a small set. We propose a novel method for training CGANs which allows us to condition on a sequence of continuous latent distributions $f^{(1)}, \ldots, f^{(K)}$. This training allows CGANs to generate samples from a sequence of distributions. We apply our method to paintings from a sequence of artistic movements, where each movement is considered to be its own distribution. Exploiting the temporal aspect of the data, a vector autoregressive (VAR) model is fitted to the means of the latent distributions that we learn, and used for one-step-ahead forecasting, to predict the latent distribution of a future art movement $f^{(K+1)}$. Realizations from this distribution can be used by the CGAN to generate 'future' paintings. In experiments, this novel methodology generates accurate predictions of the evolution of art. The training set consists of a large dataset of past paintings. While there is no agreement on exactly what current art period we find ourselves in, we test on plausible candidate sets of present art, and show that the mean distance to our predictions is small.

## 1. Introduction

Periodization in art history is the process of characterizing and understanding art 'movements'[1] and their evolution over time. Each period may last from years to decades, and encompass diverse styles. It is 'an instrument in ordering the historical objects as a continuous system in time and space' [1], and it has been the topic of much debate among art historians [2]. In this paper, we leverage the success of data generative models such as generative adversarial networks (GANs) [3] to

---

[1]Note that by art 'movements' we mean periods.

**Figure 1.** Sample of images generated by the proposed framework. These images estimate 'future' paintings.

learn the distinct features of widely agreed upon art movements, tracing and predicting their evolution over time.

Unlike previous work [4,5], in which a clustering method is validated by showing that it recovers known categories, we take existing categories as given, and propose new methods to more deeply interrogate and engage with historiographical debates in art history about the validity of these categories. Time labels are critical to our modelling approach, following what one art historian called 'a basic datum and axis of reference' in periodization: 'the irreversible order of single works located in time and space'. We take this claim to its logical conclusion, asking our method to forecast into the future. As the dataset we use covers agreed upon movements from the fifteenth to the twentieth century, the future is really our present in the twenty-first century. As it can be seen in figure 1, we are thus able to evaluate one hypothesis about what movement we find ourselves in at present, namely Post-Minimalism, by comparing the 'future' art we generate with our method to Post-Minimalist art (which was not part of our training set) and other recent movements.[2]

We consider the following setting: each observed image $x_i$ has a cluster label $k_i \in \{1, \ldots, K\}$ and resides in an image space $\mathcal{X}$, where we assume that $\mathcal{X}$ is a mixture of unknown distributions $f_X^{(1)}, \ldots, f_X^{(K)}$. For each observed image, we have $x_i \sim f_X^{(k_i)}$. We assume that, given data from the sequence of time-ordered distributions $f_X^{(1)}, \ldots, f_X^{(K)}$, it is possible to approximate the next distribution, $f_X^{(K+1)}$. For example, each $x_i$ could be a single painting in a dataset of art. Further, each painting can be associated with one of $K$ art movements such as Impressionism, Cubism or Surrealism. In this example, $f_X^{(K+1)}$ represents an art movement of the future.

In this work, we are interested in generating images from the next distribution $f_X^{(K+1)}$. However, modelling directly in the image space $\mathcal{X}$ is complicated. Therefore, we assume that there is an associated lower-dimensional latent space $\mathcal{C}$, such that each image distribution $f_X^{(k)}$ is associated with a latent distribution $f_C^{(k)}$ in $\mathcal{C}$ and every observed image $x_i$ is associated with a vector $c_i$ in the latent space which we refer to as a *code*. We chose a latent space of lower dimension than that of the image space to facilitate the modelling process: for example, if $x_i$ is an image of $128 \times 128$ pixels, $c_i$ could be a code of dimension 50. Thus, we consider the image-code-cluster tuples $(x_1, c_1, k_1), \ldots, (x_N, c_N, k_N)$.

Our contribution is as follows: we use a novel approach to conditional generative adversarial networks (CGANs, [6]) that conditions on continuous codes, which are in turn modelled with vector autoregression (VAR, [7]). The general steps of the method are:

(i) For each image $x_i$ learn a coding $c_i$; $i = 1, \ldots, N$.
(ii) Train a CGAN using $(x_1, c_1), \ldots, (x_N, c_N)$ to learn $X \mid C$.
(iii) Model latent category distributions $f_C^{(1)}, \ldots, f_C^{(K)}$.
(iv) Predict $f_C^{(K+1)}$ and draw new latent samples $c_1^*, \ldots, c_M^* \sim f_C^{(K+1)}$.
(v) Sample new images $x_j^* \sim X|C = c_j^*$ using CGAN from step 2; $j = 1, \ldots, M$.

---

[2]As the real paintings from recent movements are copyrighted, they cannot be shown here. For visual comparison, see https://github.com/cganart/gan_art_2019 to find links to the original paintings.

Typically, the aim of GANs is to generate realizations from an unknown distribution with density $f_X(x)$ based on the observations $x_1, \ldots, x_N$. Existing approaches for training GANs are mostly focused on learning a *single* underlying distribution of training data. However, this work is concerned with handling a *sequence* of densities $f_X^{(1)}, \ldots, f_X^{(K)}$. As mentioned earlier, our objective is to generate images from $f_X^{(K+1)}$ using trend information we learn from data from the previous distributions. To do this, a VAR model is used for the sequence of *latent* distributions $f_C^{(1)}, \ldots, f_C^{(K)}$.

CGANs generate new samples from the conditional distribution of the data $X$ given the latent variable $C$. The majority of current CGAN literature (e.g. [8,9]) considers the latent variable $C$ as a discrete distribution (i.e. labels) or as another image. In this work, however, the variable $C$ is a continuous random variable. Although, conditioning on discrete labels is a simple and effective way to generate images from an individual category without needing to train a separate GAN for each, discrete labels do not provide a means to generate images from an unseen category. We show that conditioning on a continuous space can indeed solve this issue.

Our CGAN is trained on samples from $K$ categories. Based on this trained CGAN, 'future' new samples $x^*$ from category $K + 1$ are obtained sampling from $X \mid C$, where $C \sim f_C^{(K+1)}$. In other words, we use a CGAN to generate images based upon the prediction given by the VAR model in the latent space, i.e. generate new images from $f_X^{(K+1)}$. In this paper, the latent representations are obtained via an autoencoder—see §2.3 later.

It is important to point out that the method does not aim to model a sequence of individual images, but a sequence of *distributions* of images. Recalling the art example: an individual painting in the Impressionism category is not part of a sequence with e.g. another individual painting in the Post-Impressionism category. It is the two categories themselves that are to be modelled as a sequence.

The novel contribution of this paper can be summarized as generating images from a distribution with 0 observations by exploiting the sequential nature of the distributions via a latent representation. This is achieved by combining existing methodologies in a novel fashion, while also exploring the seldom-used concept of a CGAN that conditions on continuous variables. We assess the performance of our method using widely agreed upon art movements from the public domain of WikiArt dataset [10] to train a model which can generate art from a predicted movement; comparisons with the real-art movements that follow the training set show that the prediction is close to ground truth.

To summarize, the overall objectives considered in this paper are:

— Derive a latent representation $c_1, \ldots, c_N$ for training sample $x_1, \ldots, x_N$.
— Find a model for the $K$ categories in this latent space.
— Predict the 'future', i.e. category $K + 1$, in the latent space.
— Generate new images that have latent representations corresponding to the $(K + 1)$th category.

There exist some methods that have used GANs and/or autoencoders for predicting new art movements. For instance, Vo & Soh [11] propose a collaborative variational autoencoder [12] that is trained to project existing art pieces into a latent space, then to generate new art pieces from imaginary art representations. However, this work generates new art subject to an auxiliary input vector to the model and does not capture sequential information across different movements. On the other hand, Sigaki *et al.* [13] proposed an alternative approach to measure the evolution of art movements on a double scale of simplicity–complexity and order–disorder, both related to the local ordinal pattern of pixels throughout the images. This latter method is, however, not used to generate new artwork, from existing movements nor future predictions. The idea of using GANs to generate new art movements has also been explored by Elgammal *et al.* [14] via creative adversarial networks. These networks were designed to generate images that are hard to categorize into existing movements. Unlike our work, there is no modelling of the sequential nature of movements.

The essential difference between our proposed model and other conditional generative models such as [11,13] is that existing work does not aim to capture the flow of influence among the several art movements to predict what is happening in the near future art movement. What they care about is how to generate new art instances based on a desired condition of users' interests. Hence, we cannot directly compare the artefacts generated by existing methods with what we aim to generate as the near future art movements. Finally, modelling the sequential nature of a dataset is not limited to images/paintings: for instance, the history of music can also be interpreted as a succession of genres. Using GANs for music has been explored by Mogren [15], but again modelling the sequential nature of genres has not been explored.

# 2. Methodology

We now describe the general method used to model a sequence of latent structures of images and use this model to make future predictions. The full procedure is outlined in algorithm 1. The remaining subsections are devoted to discussing the main steps of this algorithm in detail.

## 2.1. Generative adversarial networks

A GAN comprises two artificial neural networks: a *generator* $G$ and a *discriminator* $D$. The two components are pitted against each other in a two-player game: given a sample of real images, the generator $G$ produces random 'fake' images that are supposed to look like the real sample, while $D$ tries to determine whether these generated images are fake or real. An important point is that only $D$ has access to the sample of real images; $G$ will initially output noise, which will improve as $D$ sends feedback. At the same time, $D$ will train to become better and better at judging real from fake, until an equilibrium is reached, such that the distribution implicitly defined by the generator corresponds to the underlying distribution of the training data—see [3] for more details. In practice, the training procedure does not guarantee convergence. A good training procedure, however, can bring the distribution of the generator very close to its theoretical optimum.

CGANs [6] are an extension of GANs where the generator produces samples by conditioning on extra information. The data that we wish to condition on is fed to both the generator and discriminator. The conditioning information can be a label, an image or any other form of data. For instance, [6] generated specific digits that imitate the MNIST dataset by conditioning on a one-hot label of the desired digit.

More technically: a generator, in the GAN framework, learns a mapping $G : z \rightarrow x$ where $z$ is random noise and $x$ is a sample. A *conditional* generator, on the other hand, learns mapping $G : (z, c) \rightarrow x$, where $c$ is the information to be conditioned on. The pair $(x, c)$ is input to the discriminator as well, so that it learns to estimate the probability of observing $x$ *given* a particular $c$. The objective function of the CGAN is similar to the standard GANs: the conditional distribution of the generator converges to the underlying conditional distribution of $X \mid C$ [16].

In our setting, a CGAN is trained on a dataset of images $x_1, \ldots, x_N$ where every image $x_i$ is associated with a latent vector $c_i \in \mathbb{R}^{d_c}$. The latent vectors are considered realizations of a mixture distribution with density

$$f_C(c) = \sum_{k=1}^{K} w_k f_C^{(k)}(c), \quad \text{where} \sum_{k=1}^{K} w_k = 1. \tag{2.1}$$

Each density $f_C^{(k)}$ corresponds to an artistic movement, and $f_C^{(1)}, \ldots, f_C^{(K)}$ is considered to be a sequence of densities. We again stress that $C$ is assumed to be a *continuous* random variable. See §2.2 for details of how to train CGANs with a continuous latent space.

The conditional generator is trained to imitate images from density $f_{X \mid C}(x \mid c)$. After being trained, the generator can be used to sample new images. This can be achieved by sampling from the latent space $\mathcal{C}$. Note that we are capable of sampling from areas of $\mathcal{C}$ where few data are observed during training. Then the generator is forced to condition on 'new' information, thus producing images with novel features.

## 2.2. Continuous CGAN: training details

Usually, CGANs condition on a discrete label [6] and are straightforward to train: training sets for this task contain many images for each label category. Then training $G$ and $D$ on generated images is a two-step task: (i) pick a label $c$ randomly and generate image $x$ given this label, then (ii) update model parameters based on the $(x, c)$ pair.

When training a continuous CGAN, however, each $x_i$ in the training set is associated with a unique $c_i$. Picking an existing $c_i$ to generate a new $x$ is an unsatisfactory solution: if done during training, $G$ would learn to generate exact copies of the original $x_i$ associated with $c_i$. We would also lose the flexibility of being able to use the whole continuous latent space, instead selecting individual points in it.

As mentioned in §2.1, the latent vectors $c_1, \ldots, c_N$ are considered realizations of mixture distribution $f_C$ with components $f_C^{(1)}, \ldots, f_C^{(K)}$ and weights $w_1, \ldots, w_K$. We propose the novel idea of approximating the latent distribution as a mixture of multivariate normals, and of using this approximation to sample new $c^*$ during and after training. We compute the sample means and covariances $(\hat{\boldsymbol{\mu}}_C^{(1)}, \hat{\Sigma}_C^{(1)}) \ldots, (\hat{\boldsymbol{\mu}}_C^{(K)}, \hat{\Sigma}_C^{(K)})$. Then each density component $f_C^{(k)}$ is approximated as $N(\hat{\boldsymbol{\mu}}_C^{(k)}, \hat{\Sigma}_C^{(k)})$. The weights $w_k$ are estimated as $\hat{w}_k$, the proportion of training images in category $k$.

Generating new $x$ for the purpose of training, or for producing images in a trained model, is then done by (i) picking category $k$ with probability $\hat{w}_k$, (ii) drawing a random $c \sim N(\hat{\boldsymbol{\mu}}_C^{(k)}, \hat{\Sigma}_C^{(k)})$, and (iii) using the generator with the current parameters to produce $x \mid c$.

Note that by assuming a fixed (Gaussian) form for the conditional distributions, we are appealing to the same sort of (Laplace) assumption that underpins variational Bayes. This speaks to the possibility of using approximate Bayesian (i.e. variational) inference to describe, or indeed implement, the current scheme.

## 2.3. Obtaining the latent codes via autoencoders

So far we have assumed that each image $x_i$ is associated with a *latent vector* $c_i \in \mathbb{R}^{d_c}$. In principle, these latent representations of the images can be obtained with any method. Some reasonable properties of the method are as follows:

— If images $x_i$ and $x_j$ are similar, then their associated latent vectors $c_i$ and $c_j$ should be close. Here the concept of closeness or 'similarity' is not restricted to the the simple pixel-wise norm $\|x_i - x_j\|_2^2$, but is instead a broader concept of similarity between the features of the images. For instance, two images containing boats should be close in the latent space even if the boat is in a different position in each image.
— Sampling from $f_C(c)$ needs to be straightforward.

Autoencoders are an easy and flexible choice that satisfies the two points above. For this reason, we choose to use autoencoders in this work. However, we stress that any method with the properties described in the list above can be used to obtained the latent codes. An alternative choice could be the method used by Sigaki *et al*. [13] to measure artwork on two scales of order–disorder and complexity–simplicity.

Johnson *et al*. [17] made use of a *perceptual loss* function between two images to fulfil the tasks of style transfer and super-resolution. The method, which builds on earlier work by Gatys *et al*. [18], is based on comparing high-level features of the images instead of comparing the images themselves. The high-level features are extracted via an auxiliary pre-trained network, e.g. a VGG classifier [19]. The same concept can be applied to autoencoders, and the resulting latent space satisfies the above point about preservation of image similarity. We use this perceptual loss specifically for art data: the details are in §3.1.

Note that the latent space is learned without knowledge of categories $k = 1, \ldots, K$. It is assumed that, when moving from $\mathcal{X}$ to $\mathcal{C}$, the $K$ distributions $f_C^{(1)}, \ldots, f_C^{(K)}$ are somewhat ordered. This is, however, not guaranteed. The assumption can be easily tested, as it is done in §3.2.

## 2.4. Predicting the future latent distribution

We make the assumption that $f_X^{(1)}, \ldots, f_X^{(K)}$ have a non-trivial relationship, and that they can be interpreted as being a 'sequence of distributions'. Furthermore, we assume that this sequential relationship is preserved when we map the distributions to $f_C^{(1)}, \ldots, f_C^{(K)}$ using the autoencoder. The key part of our method is that we assume the latent space and latent distributions to be simple enough that we can predict $f_C^{(K+1)}$, which is completely unobserved. Then we aim to use the same conditional generator trained as described in §2.1 to sample from $f_X^{(K+1)}$, which is also unobserved. In our setting, the sequence of densities $f_C^{(1)}, \ldots, f_C^{(K)}$ represents, in the case of the WikiArt dataset, a latent sequence of artistic movements.

The underlying distribution of $f_C^{(1)}, \ldots, f_C^{(K)}$ is unknown. Suppose we have realizations from each of these distributions (see §2.3); then we model the sequence of latent distributions as follows. We assume that each $f_C^{(k)}$ follows a normal distribution $N(\boldsymbol{\mu}_C^{(k)}, \Sigma_C^{(k)})$. Denote $\hat{\boldsymbol{\mu}}_C^{(k)}$, an estimator of $\boldsymbol{\mu}_C^{(k)}$, as the sample mean of $f_C^{(k)}$. Then the mean is modelled using the following vector autoregression (VAR) model with a linear trend term:

$$\hat{\boldsymbol{\mu}}_C^{(k)} = \boldsymbol{\alpha} + k\boldsymbol{\beta} + A\hat{\boldsymbol{\mu}}_C^{(k-1)} + \boldsymbol{\epsilon}, \quad \boldsymbol{\epsilon} \sim N(\mathbf{0}, \Sigma_\epsilon). \tag{2.2}$$

Vectors $\boldsymbol{\alpha}$, $\boldsymbol{\beta}$ and matrices $A$, $\Sigma_\epsilon$ are parameters that need to be estimated. Estimation is performed using a sparse specification (e.g. via LASSO) in the high-dimensional case.

Once the parameters are estimated we can predict $\hat{\boldsymbol{\mu}}_C^{(K+1)}$, the latent mean of the unobserved future distribution.

The covariance of $f_C^{(K+1)}$ is estimated by $\hat{\Sigma}_C^{(K+1)} = \frac{1}{K}(\hat{\Sigma}_C^{(1)} + \cdots + \hat{\Sigma}_C^{(K)})$. For the WikiArt dataset we observed little change in the empirical covariance structure of $f_C^{(1)}, \ldots, f_C^{(K)}$, and therefore elected to use an average of the observed covariances.

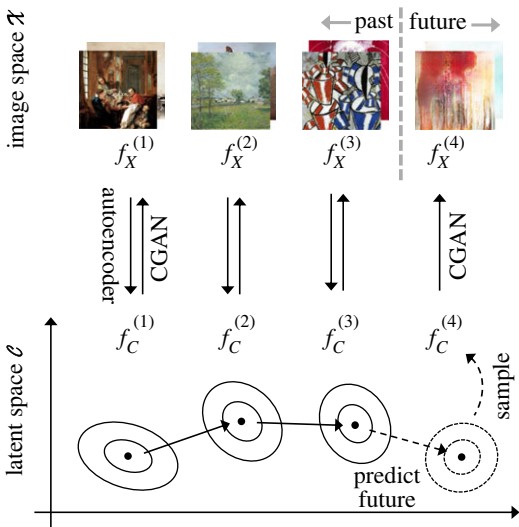

**Figure 2.** Diagram illustrating our method. The latent space $\mathcal{C}$ is chosen to be lower-dimensional than the image space $\mathcal{X}$. Moving from $\mathcal{X}$ to $\mathcal{C}$ does not necessarily need to be done via an autoencoder, as noted in §2.3 (images from public domain of [10]).

---

**Algorithm 1.** Predicting using CGANs.

---

1: Train a CGAN with generator $G(z|c)$ and discriminator $D(x|c)$ on real and fake pairs $\{x_i, c_i\}$.

2: Estimate $\hat{\mu}_C^{(1)}, \ldots, \hat{\mu}_C^{(K)}$, the sample means of the $K$ categories of latent codes.

3: Fit a VAR model on $\hat{\mu}_C^{(1)}, \ldots, \hat{\mu}_C^{(K)}$ and predict $\hat{\mu}_C^{(K+1)}$.

4: Draw 'future' code $c^* \sim N(\hat{\mu}_C^{(K+1)}, \hat{\Sigma}_C^{(K+1)})$, where $\hat{\Sigma}_C^{(K+1)} = \frac{1}{K}(\hat{\Sigma}_C^{(1)} + \cdots + \hat{\Sigma}_C^{(K)})$.

5: Generate new images by sampling from $G(z|c^*)$.

---

The future latent distribution $f_C^{(K+1)}$ is therefore approximated as $N(\hat{\mu}_C^{(K+1)}, \hat{\Sigma}_C^{(K+1)})$.

The entire method described in §2 is outlined in algorithm 1.

## 2.5. Theoretical notes on the procedure

The autoencoder, or any alternative method that satisfies the properties laid out in §2.3, maps each image $x_i$ to a low-dimensional latent vector $c_i$. This mapping implicitly defines a distribution in the latent space, and our assumption is that each distribution $f_X^{(k)}$ of images is mapped to a distribution $f_C^{(k)}$ in the latent space.

The conditional generator produces samples from distribution $f_{X|C}^G$, where the latent code $c$ can come from any of the latent distributions $f_C^{(k)}$, $k = 1, \ldots, K$. Note the superscript '$G$' in $f_{X|C}^G$, indicating that the distribution implicitly defined by the generator does not necessarily equal the theoretical training optimum $f_{X|C}$ (as mentioned in §2.1). Nevertheless, we will proceed under the assumption that a good training procedure results in a conditional generator close to the theoretical equilibrium. The conditional generator, just like the autoencoder, does not know which movement $x$ and $c$ belong to.

Recall that the overall distribution of all latent codes was modelled as a mixture of the $K$ movement-wise distributions in equation (2.1). Our method is based on the premise that, while the conditional GAN is trained on the whole space of the $K$ movements, new samples can be generated from an individual movement $f_X^{(k)}$ by conditioning on random variable $C$ from $f_C^{(k)}$. That is, if we draw $c_1, \ldots, c_m \sim f_C^{(k)}$, the conditional generator will produce sample $x_1, \ldots, x_m$ whose empirical distribution is close to $f_X^{(k)}$. This is motivated by marginalizing $X$ out of $f_{X|C}^G(x|c)$:

$$\int_c f_{X|C}^G(x|c)\, \mathrm{d}F_C^{(k)} = f_X^{(k)}(x), \tag{2.3}$$

**Table 1.** Summary of the WikiArt dataset. 'Year' is the approximate median year of the art movement, $n$ is number of images.

| movement | year | $n$ | movement | year | $n$ |
|---|---|---|---|---|---|
| Early Renaissance | 1440 | 1194 | Fauvism | 1905 | 680 |
| High Renaissance | 1510 | 1005 | Expressionism | 1910 | 6232 |
| Mannerism | 1560 | 1204 | Cubism | 1910 | 1567 |
| Baroque | 1660 | 3883 | Surrealism | 1930 | 3705 |
| Rococo | 1740 | 2108 | Abstract Expressionism | 1945 | 1919 |
| Neoclassicism | 1800 | 1473 | Tachisme/Art Informel | 1955 | 1664 |
| Romanticism | 1825 | 7073 | Lyrical Abstraction | 1960 | 652 |
| Realism | 1860 | 8680 | Hard Edge Painting | 1965 | 362 |
| Impressionism | 1885 | 8929 | Op Art | 1965 | 480 |
| Post-Impressionism | 1900 | 5110 | Minimalism | 1970 | 446 |

where $F$ is the cumulative distribution function associated with $f$, and $f_{X|C}^G$ is the distribution implicitly defined by the generator. The overall procedure is illustrated in figure 2.

# 3. Results

The performance of our method presented in §2 is demonstrated on the public domain of WikiArt dataset,[3] where each category represents an art movement. All experiments are implemented with Tensorflow [20] via Keras, and run on a NVIDIA GeForce GTX 1050.[4]

After the introduction of the setting, the structure of the resulting latent spaces is discussed in §3.2. Finally, §3.3 describes the prediction and generation of future art from $f_X^{(K+1)}$.

## 3.1. WikiArt results

The dataset considered is the publicly available WikiArt dataset, which contains 103 250 images categorized into various movements, types (e.g. portrait or landscape), artists and sometimes years. We use the central square of each image, re-sized to $128 \times 128$ pixels. Note that a small number of raw images are unable to be reshaped into our desired format, reducing the total sample size to 102 182.

Additionally, note that all images considered are paintings; images that are tagged as 'sketch and study', 'illustration', 'design' or 'interior' were excluded. The remaining images can then be categorized into 20 notable and well-defined artistic movement from Western art history (table 1).

In order to apply algorithm 1, each image $x_i$ in the dataset needs to be associated with a latent vector $c_i$. As described in §2.3, a non-variational autoencoder with *perceptual loss* is utilized. Note again that the category labels associated with each image are not revealed to the autoencoder when training it. Two autoencoders are separately trained with *content* loss and *style* loss which are now defined:

*Content loss*

$$\mathcal{L}_{\text{content}}(x_a, x_c) = \frac{1}{C_j H_j W_j} \|\phi_j(x_a) - \phi_j(x_c)\|_2^2,$$

where $\phi_j(\cdot)$ is the $j$th convolutional activation of a trained auxiliary classifier, while $C_j$, $H_j$, $W_j$ are the dimensions of the output of that same $j$th layer. Johnson *et al*. [17] and Gatys *et al*. [18] explain how this loss function is minimized when two images share extracted features that represent the overall shapes and structures of objects and backgrounds; they also discuss how the choice of $j$ influences the result. Each image $x$ is thus associated with a content latent vector $c_c$.

---

[3]See https://www.wikiart.org/.

[4]All the code is available at https://github.com/cganart/gan_art_2019.

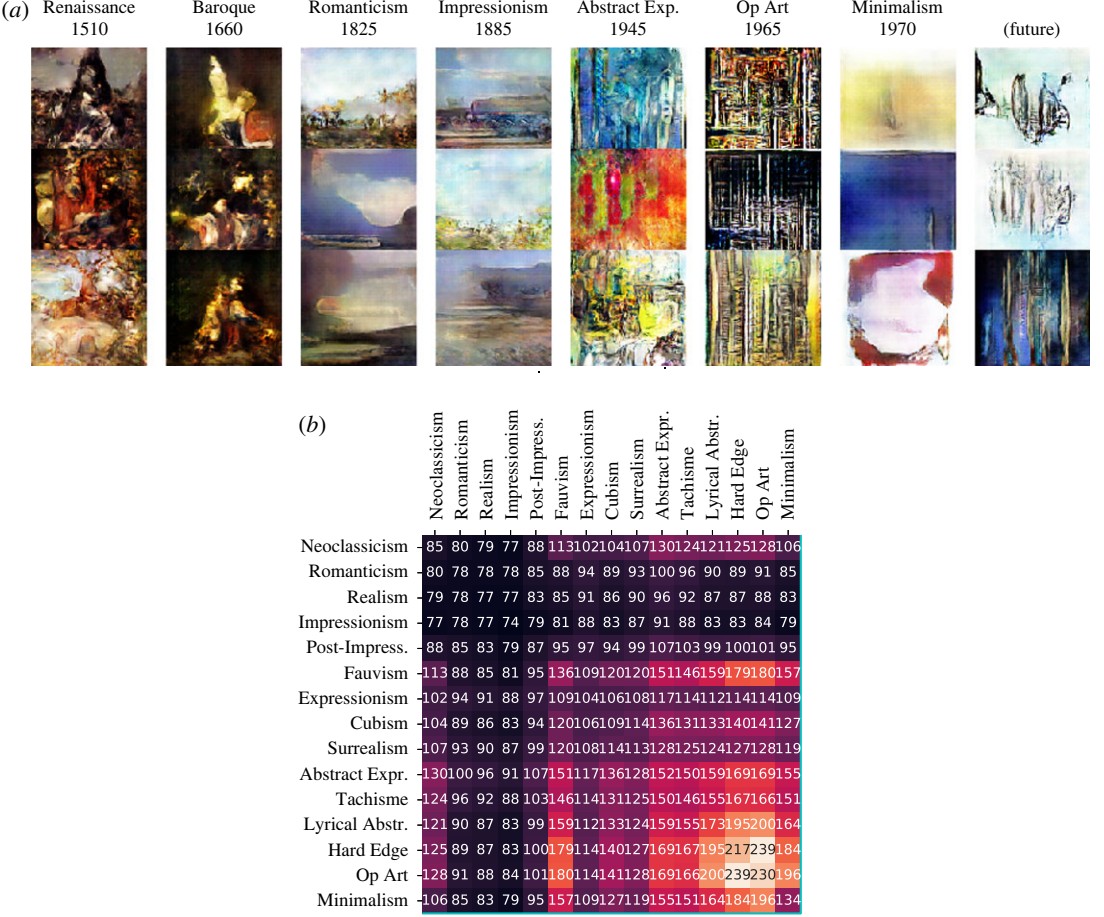

**Figure 3.** (*a*) Evolution of artistic movements as generated by our method. The last column, labelled 'future', is the prediction $x_1^*, \ldots, x_M^* \sim f_X^{(K+1)}$, where $K = 20$. Note that there is no relationship between images in the same row but different column (images from public domain of [10]). (*b*) Matrix of within-cluster variance (diagonal elements) and between-cluster variance (off-diagonal elements) for all pairs of artistic movements. The values are averaged across all dimensions of the latent space.

*Style loss*

$$\mathcal{L}_{\text{style}}(x_a, x_c) = \frac{1}{C_j H_j W_j} \|G_j(x_a) - G_j(x_c)\|_2^2,$$

where $G_j(\cdot)$ is the Gram matrix of layer $j$ of the same auxiliary classifier used for the content loss. This loss function is used to measure the similarity between images that share the same repeated textures and colours, which we collectively call *style*. Each image $x$ is thus associated with a style latent vector $c_s$.

The auxiliary classifier is obtained by training a simplified version of the VGG16 network [19] on the tinyImageNet dataset.[5] The VGG classifier is simplified by removing the last block of three convolutional layers, thus adapting the architecture to $128 \times 128$ images rather than $256 \times 256$. Once each image $x$ has its content and style latent vectors, these are concatenated to obtain $c = [c_s^T, c_c^T]^T$.

Finally, the CGAN is trained by conditioning on the continuous latent space, as described in algorithm 1. Details about network architecture and training can be found in appendix A. Figure 3 contains examples of generated images from various artistic movements, together with a quantitative assessment of within- and between-movement average latent variance. Some qualitative comments can be remarked (whereas quantitative evaluations are in §§3.2 and 3.3):

— There is very good between-movement variation and within-movement variation. It is hard to find two generated images that are similar to each other.

[5]See https://tiny-imagenet.herokuapp.com/.

**Table 2.** Performance when regressing movement label on movement-means of various types of latent spaces.

|        | standard | style | content | joint | concat. |
|--------|----------|-------|---------|-------|---------|
| $R^2$  | 0.19     | 0.41  | 0.24    | 0.39  | 0.41    |
| cor.   | 0.19     | 0.24  | 0.12    | 0.22  | 0.20    |

— One of the main reasons that guided the use of a perceptual autoencoder was the fact that movements vary not only in style (e.g. colour, texture) but also in content (e.g. portrait or landscape). From this point of view our method is a success. Each movement appears to have its own set of colours and textures. Additionally, movements that were overwhelmingly portraits in the training set (e.g. Baroque) result in generated images that mostly mimic the general structure of human figures. Similarly, movements with a lot of landscapes (e.g. Impressionism) result in generated images that are also mostly landscapes; the latter tend to be of very good quality.

— More abstract movements (e.g. Lyrical Abstraction) result in very colourful generated images with little to no structure, as is to be expected. Interesting behaviours can be observed: Op Art paintings, for instance, are generally very geometric and often remind of chessboards, and the generator's effort to reproduce this can be clearly observed (figure 3). The same can be said of Minimalist art, where many paintings are monochromatic canvas; the generator does a fairly good job at reproducing this as well.

A drawback of using the WikiArt dataset is that the relatively small number of movements ($K = 20$) forces the use of a very sparse version of VAR [21]. As a result, the predicted future mean $\hat{\boldsymbol{\mu}}_C^{(K+1)}$ is almost entirely determined by the linear trend component of the VAR model, $\boldsymbol{\alpha} + k\boldsymbol{\beta}$; the autoregressive component $A\hat{\boldsymbol{\mu}}_C^{(K)}$ is largely non-influential, as the parameter matrix $A$ is shrunk to 0 by the sparse formulation.

## 3.2. Latent space analysis

Section 2.3 mentioned that it is not guaranteed that the $K$ categories will actually be ordered in the latent space, although it is expected. We implement a simple heuristic to test this in the WikiArt case: suppose that $\boldsymbol{y} = [1, \ldots, K]^T$ and that $M \in \mathbb{R}^{K \times d_c}$ is a matrix with $\hat{\boldsymbol{\mu}}_C^{(1)}, \ldots, \hat{\boldsymbol{\mu}}_C^{(K)}$ as rows ($d_c$ is the dimension of the latent space). Then we can fit a simple linear regression $\boldsymbol{y} = M\boldsymbol{\beta} + \boldsymbol{\epsilon}$, where $\boldsymbol{\epsilon} \sim N(\boldsymbol{0}, \sigma^2 I)$ and $\boldsymbol{\beta}$ and $\sigma^2$ are parameters. We do this for various types of latent vectors obtained with different loss functions: pixel-wise cross-entropy, style-only, content-only, the sum of the latter two (joint), and a concatenation of style-only and content-only.

Table 2 displays the $R^2$ values (the coefficient of determination) for each type of latent vector, which can be directly compared, as matrix $M$ always has size $K \times d_c$. The mean of the absolute correlations between pairs of the 100 dimensions of each latent space is also presented in table 2. This is a simple measure of how the various dimensions of the latent vectors are correlated with each other.

The results suggest using a perceptual loss instead of a pixel-wise loss: the results for the last four columns (the different types of perceptual losses) are much better than the 'standard' latent space obtained via pixel-wise cross-entropy. Further, the results suggest using two separate autoencoders for style and loss, and then concatenating the resulting latent vectors: the last column has the highest $R^2$ of all five methods, while also having a between-dimensions correlation that is lower than using a sum of style loss and content loss. Overall, this is an impressive result: recall that the autoencoders do not have access to the movement labels $k \in \{1, \ldots, K\}$. Despite this, the latent vectors are able to predict those same movement labels quite accurately. This result confirms that there is indeed a natural ordering of the art movements (which corresponds to their temporal order), and that this natural ordering is reflected in the latent space. This can also be seen in the means of the clusters of latent vectors in figure 4.

Figure 5 displays a heatmap of distances between pairs of movements in the latent space. Most notably, the matrix exhibits a block-diagonal structure. This means that (i) movements that are chronologically close are also close in the latent space, and (ii) there tends to be an alternation between series of movements being similar to each other and points where a new movement breaks from the past more significantly. Figure 5 also shows the position of predicted and real 'future' (or current) movements relative to the movements in the training set. More detail can be found is §3.3.

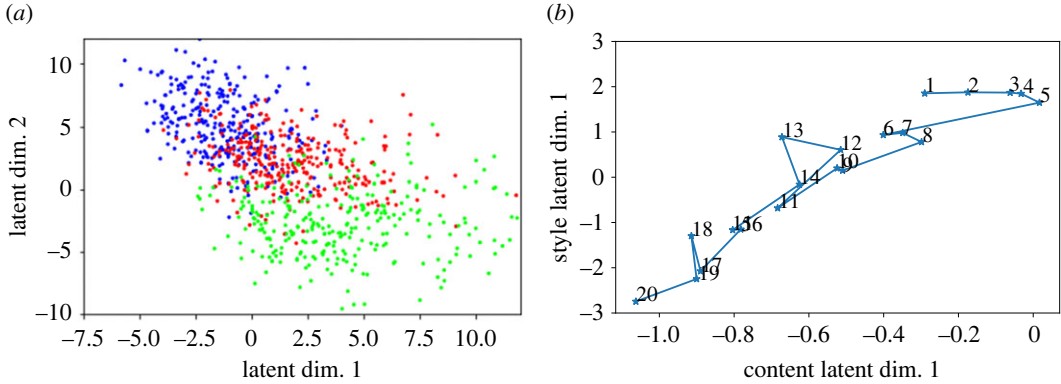

**Figure 4.** (*a*) Visualization of the latent space. The two dimensions of **C** that best correlate with movement index were chosen, and plotted in the *x*- and *y*-axes. Three movements are highlighted: Early Renaissance (blue, top-left), Impressionism (red, centre), and Minimalism (green, bottom-right). The clustering and temporal ordering are clearly visible. (*b*) Means of all 20 training art movements (table 1) visualized to emphasize the temporal progression. The first PCA component of content and style latent spaces are shown in the *x*- and *y*-axes, respectively.

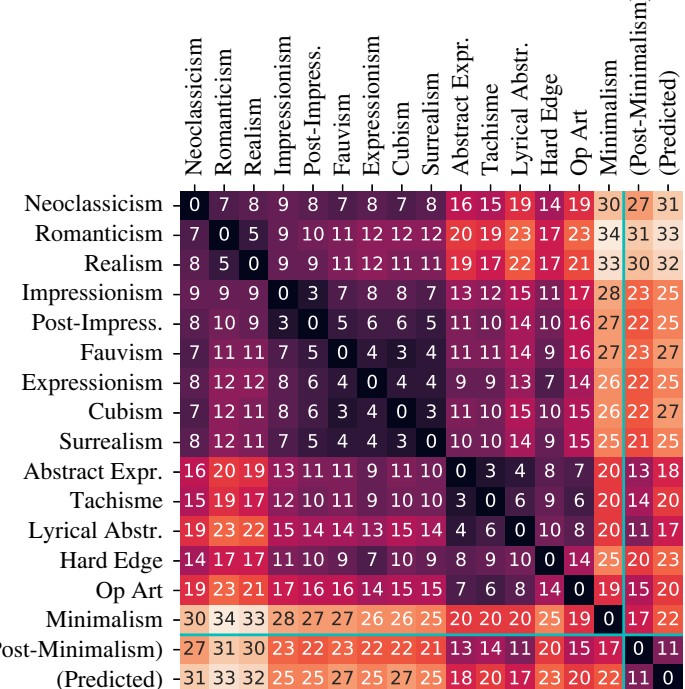

**Figure 5.** Matrix of Euclidean distances between the means of individual movements in the latent space. The movements are ordered chronologically. The last two columns/rows represent true Post-Minimalist paintings as well as our prediction. Note the block-diagonal tendencies.

## 3.3. Future prediction

Once the CGAN is fully trained on the dataset of $K$ training set categories, autoregression methods are used to generate from the unobserved $(K+1)$th category (the future). As described in §2.4, we use a simple linear trend plus sparse VAR on the means $\hat{\boldsymbol{\mu}}_C^{(1)}, \ldots, \hat{\boldsymbol{\mu}}_C^{(K)}$ of the $K$ categories in the latent space. This results in predicted mean $\hat{\boldsymbol{\mu}}_C^{(K+1)}$, while the predicted covariance $\hat{\Sigma}_C^{(K+1)}$ is simply the mean of the $K$ training covariances. Then we sample new latent vectors from $N(\hat{\boldsymbol{\mu}}_C^{(K+1)}, \hat{\Sigma}_C^{(K+1)})$, and feed them to the trained conditional generator together with the random noise vector. The result is generated images that condition on an area of the latent space which is not covered by any of the

**Table 3.** Two types of distances between real recent movements (columns) and either future predictions ($K + 1$) or last movement in the training set (Minimalism, $K$).

| | Post-Minimalism | New Casualism |
|---|---|---|
| Euclid.($K + 1$) | 11.8 | 12.3 |
| Euclid.($K$) | 22.8 | 21.0 |
| MMD($K + 1$) | 0.15 | 0.18 |
| MMD($K$) | 0.27 | 0.25 |

existing movements. Instead, this latent area is placed in a 'natural' position after the sequence of $K$ successive movements. A collection of generated 'future' images can be found in figure 3.

As summarized in table 1, the WikiArt dataset only contains large, well-defined art movements up to the 1970s, the most recent one being Minimalism. The same dataset, however, also contains smaller movements that were developed after Minimalism. In particular, Post-Minimalism and New Casualism can be considered successors of the latest of the $K = 20$ training movements, but they contain too few images to be considered for training the CGAN. They can, however, be used to compare our 'future' predictions with what actually came after the last movement in the training set. We use the same autoencoder to map each image in Post-Minimalism and New Casualism. Then, after generating images from predicted movement $f_C^{(K+1)}$, we compute the Euclidean distance of the means and MMD distance [22] from the real small movements in the latent space. The results are summarized in table 3 and are included in the distance matrix in figure 5.

The results indicate a success: according to all metrics, the distance between the generated future and the real movements is small when compared with other between-movement distances shown in figure 5. In particular, the generated images are closer to Post-Minimalism and New Casualism than they are to the last training movement, i.e. Minimalism. This indicates that our prediction of the future of art is not a mere copy of the most recent observed movement, but rather a jump in the right direction towards the true evolution of new artistic movements.

This positive result can be contrasted with a simpler approach described in appendix B, where a standard autoencoder is used for both latent modelling and generation of new images.

## 4. Discussion

In this paper, we introduced a novel machine learning method to bring new insights to the problem of periodization in art history. Our method is able to model art movements using a simple low-dimensional latent structure and generate new images using CGANs. By reducing the problem of generating realistic images from a complicated, high-dimensional image space to that of generating from low-dimensional Gaussian distributions, we are able to perform statistical analysis, including one-step-ahead forecasting, of periods in art history by modelling the low-dimensional space with a vector autoregressive model. The images we produced resemble real art, including real art from held-out 'future' movements.

A number of modifications could be applied to the method. For instance, the learning architecture could be directly extended to predict art movements in the reverse direction, namely towards the past, when the time ordering of the input is reversed.

The method described in this paper can be applied outside of the context of art. For instance, appendix C describes the generation of photos of human faces from different years. The temporal succession of years is treated in the same manner as the temporal succession of art movements in the main body of this paper.

Data accessibility. Data are available at https://github.com/cganart/gan_art_2019 and at the Dryad Digital Repository at https://doi.org/10.5061/dryad.90cj2pq [24].
Authors' contributions. E.L. developed the methods, undertook the implementation, and drafted the manuscript. M.M. contributed to designing and implementing the experiments with CGAN and Autoencoder, participated in the data analysis, and helped in writing and draft the manuscript. H.H. contributed to the research conception and overall direction. F.D.-H.L. contributed to the research conception and overall direction. S.F. contributed to the research conception and overall direction.
Competing interests. We declare we have no competing interest.
Funding. We received no funding for this study.

# Appendix A. Neural network architecture

Tables 4 and 5 describe in more detail the architecture used in the conditional generator and conditional discriminator (respectively) of the CGAN.

*Generator architecture*

**Table 4.** Architecture of CGAN conditional generator.

| layer | activation | output dim. |
|---|---|---|
| noise input | | 100 |
| latent input | | 100 |
| concatenate inputs | | 200 |
| fully connected and reshape | ReLu | $1024 \times 4 \times 4$ |
| fractionally strided conv. ($5 \times 5$ filter) | ReLu | $512 \times 8 \times 8$ |
| fractionally strided conv. ($5 \times 5$ filter) | ReLu | $256 \times 16 \times 16$ |
| fractionally strided conv. ($5 \times 5$ filter) | ReLu | $128 \times 32 \times 32$ |
| fractionally strided conv. ($5 \times 5$ filter) | ReLu | $64 \times 64 \times 64$ |
| fractionally strided conv. ($5 \times 5$ filter) | tanh | $3 \times 128 \times 128$ |

*Discriminator architecture*

**Table 5.** Architecture of CGAN conditional discriminator.

| layer | activation | output dim. |
|---|---|---|
| image input | | $3 \times 128 \times 128$ |
| convolution ($5 \times 5$ filter, dropout) | ReLu | $64 \times 64 \times 64$ |
| convolution ($5 \times 5$ filter, dropout) | ReLu | $128 \times 32 \times 32$ |
| convolution ($5 \times 5$ filter, dropout) | ReLu | $256 \times 16 \times 16$ |
| convolution ($5 \times 5$ filter, dropout) | ReLu | $512 \times 8 \times 8$ |
| reshape and fully connected (dropout) | ReLu | 256 |
| concatenate with latent input | | 356 |
| fully connected (dropout) | ReLu | 256 |
| fully connected (dropout) | Sigmoid | 1 |

# Appendix B. Comparison with simple autoencoder

Section 3.1 described how the use of *content* and *style* losses enables the conditional GAN to model the temporal evolution of different elements of paintings. The availability of the two latent spaces (content and style) was among the justifications for using a conditional GAN for the generative process, as opposed to directly using the same autoencoder that models the latent space.

**Table 6.** Two types of distances between real recent movements (columns) and either future predictions ($K + 1$) or last movement in the training set (Minimalism, $K$).

| | Post-Minimalism | New Casualism |
|---|---|---|
| Euclid.($K + 1$) | 25.0 | 26.1 |
| Euclid.($K$) | 24.8 | 24.2 |
| MMD($K + 1$) | 0.30 | 0.35 |
| MMD($K$) | 0.31 | 0.30 |

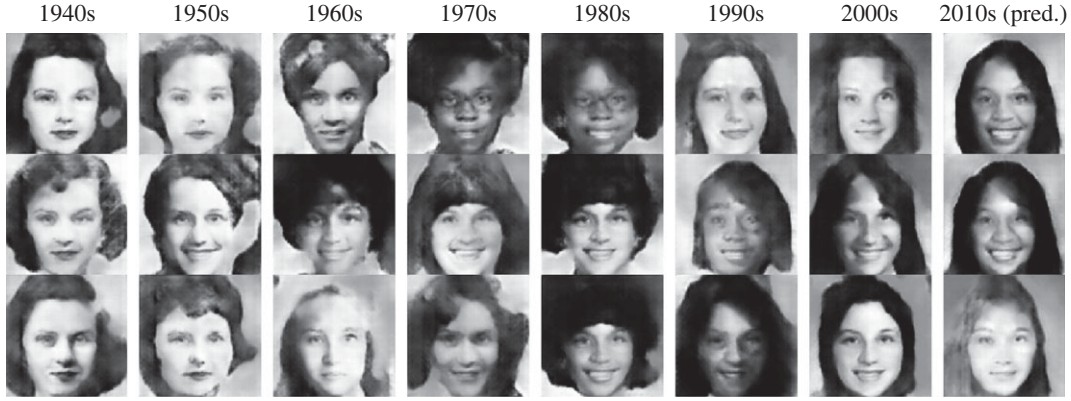

1940s    1950s    1960s    1970s    1980s    1990s    2000s    2010s (pred.)

**Figure 6.** Evolution of yearbook faces over various decades. The last column contains a prediction from the model trained on all previous decades. The prediction is a mix of images from some of the predicted distributions $f_X^{(K+1)}, \ldots, f_X^{(K+5)}$. Note that there is no relationship between images in the same row but different column (images from [23]).

A comparison can be performed between the model described in §2 and a simple model where a standard autoencoder is used both for modelling the latent space and for generating images (including estimated *future* after fitting the VAR framework to the latent space, as described in §3.3).

The main results via *content* and *style* losses (table 3) showed that the distance between predicted future and 'actual' (out of training set) future are relatively small, compared to a similar distance between consecutive recent real art movements. However, table 6 shows that this is not the case for the simple autoencoder method described in this section.

## Appendix C. Yearbook results

The yearbook dataset introduced by Ginosar *et al*. [23] contains photographs of faces of 17 163 male students and 20 248 female students from US universities. Each photo is labelled by the year it was taken, where the oldest images are from 1905 while the latest are from 2013. Post-2010 pictures are kept out of the training set, since they are going to be used as ground truth when comparing with our prediction of the future.

The model summarized in algorithm 1 is applied to this dataset. However, unlike the WikiArt example, a standard autoencoder is used to learn the latent space; the autoencoder is trained on the male images and used to predict the latent codes of the female images, and vice versa. A conditional GAN is then trained on the pairs of images and latent codes.

Although smaller than WikiArt, this dataset has the advantage of having well-defined 'year' labels, as opposed to an ordinal succession of artistic movements. The number of years covered, being more than 100, also provides benefits when fitting the VAR model in the latent space.

A collection of generated images is presented in figure 6. A few qualitative comments can be made:

— As the years progress, various changes can be noticed. Most prominently we observe the evolution of hairstyles and makeup, the diversification of race, and the increasing prevalence of smiles.
— The model is able to capture the fact that images in older years are more uniform (e.g. same hairstyles and expressions) while more recent periods show more variety.

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
