## [Reviewer comments · Royal Society Open Science]

Review History

RSOS-191569.R0 (Original submission)

Review form: Reviewer 1 (Karl Friston)

Is the manuscript scientifically sound in its present form?

Yes

Are the interpretations and conclusions justified by the results?

Yes

Is the language acceptable?

Yes

Do you have any ethical concerns with this paper?

No

Have you any concerns about statistical analyses in this paper?

No

Recommendation?

Accept with minor revision (please list in comments)

Comments to the Author(s)

I enjoyed reading this clearly described and entertaining account of "movements" in art. I thought that the description of your conditional generative adversarial networks was excellent and that you supplied the right level of detail. Clearly, there are many ways in which you could take this work. You might want to speculate about other applications (e.g., weather forecasting or medical prognosis) in your concluding paragraph. You can then point people to the appendix (for another domain of application). I have a few comments that might improve the presentation of your work. Perhaps you could consider the following:

On page 4 (line 32) did you mean "by conditioning on $f_C^{(K+1)}$) as opposed to "sampling from".

Page 4 line 36. Please say "via an automated"

Page 5 (line 9). Replace "and alternative" with "an alternative"

Page 6 (line 33) it might be useful to add something like:

"Note that assuming a fixed (Gaussian) form for the conditional distributions, we are appealing to the same sort of (Laplace) assumption that underwrites variational Bayes. This speaks to the possibility of using approximate Bayesian (i.e. variational) inference to describe (or indeed implement) the current scheme."

On page 8, it might be useful to draw a vertical line on Figure 2 and label the left as the "past" and the right as the "future".

On page 11 (line 19). I was expecting to see a characterisation of the trajectory through the C space. This would be simple to do by plotting the means of each "movement" in the principal eigenvectors of C. This would illustrate the linear progression over successive historical "movements". Perhaps you could think about adding a small figure or inset along these lines. In other words, supplement or replace Figure 4 using the eigenvectors of the latent dimensions and the centroid of the K mixtures.

Throughout the paper, it would be useful to put "movement" in inverted commas. This is because when you talk about trajectories and artistic movements, it is easy to misread this as an artist "moving" her paintbrush. When you first mention movements, emphasise that these mean "periods" or artistic styles.

I hope that these comments help should any revision be required.

Review form: Reviewer 2

Is the manuscript scientifically sound in its present form?

Yes

Are the interpretations and conclusions justified by the results?

Yes

Is the language acceptable?

Yes

Do you have any ethical concerns with this paper?

No

Have you any concerns about statistical analyses in this paper?

No

Recommendation?

Major revision is needed (please make suggestions in comments)

Comments to the Author(s)

I believe this a worthwhile contribution to neural based modelling of arts movements, and a nicely presented article which I'd recommend to be published in the journal should following issues be addressed.

A major missing piece is the lack of presentation of results from any of the related works. I think artificially generated images from collaborative variational autoencoder given the required auxiliary input vector can be compared to the results of your method.

Between-movement and within-movement variation in the experiment presented in Fig.3 can be also quantified statistically (e.g. using entropy related metrics), and not just shown visually. I think Table 1 should be compressed, and more details on the used neural network architecture should be added (as well as presentation and argumentation for simplifications on VGG16 network). Moreover, there are missing references at few points, e.g. line 27 "Unlike previous work in machine learning...".

Finally, it worths exploring whether your learning architecture can be directly extended to model arts movement in the reverse direction, namely towards the past, when time ordering of the input is inverted.

Decision letter (RSOS-191569.R0)

27-Jan-2020

Dear Mr Lisi,

The editors assigned to your paper ("Modeling and Forecasting Art Movements with CGANs") have now received comments from reviewers. We would like you to revise your paper in accordance with the referee and Associate Editor suggestions which can be found below (not including confidential reports to the Editor). Please note this decision does not guarantee eventual acceptance.

Please submit a copy of your revised paper before 19-Feb-2020. Please note that the revision deadline will expire at 00.00am on this date. If we do not hear from you within this time then it will be assumed that the paper has been withdrawn. In exceptional circumstances, extensions may be possible if agreed with the Editorial Office in advance. We do not allow multiple rounds of revision so we urge you to make every effort to fully address all of the comments at this stage. If deemed necessary by the Editors, your manuscript will be sent back to one or more of the original reviewers for assessment. If the original reviewers are not available, we may invite new reviewers.

When submitting your revised manuscript, you must respond to the comments made by the referees and upload a file "Response to Referees" in "Section 6 - File Upload". Please use this to

document how you have responded to the comments, and the adjustments you have made. In order to expedite the processing of the revised manuscript, please be as specific as possible in your response.

- Data accessibility

If you wish to submit your supporting data or code to Dryad (<http://datadryad.org/>), or modify your current submission to dryad, please use the following link:
<http://datadryad.org/submit?journalID=RSOS&manu=RSOS-191569>

- Competing interests

- Authors' contributions

- Acknowledgements

- Funding statement

on behalf of Dr Cecilia Mascolo (Associate Editor) and Marta Kwiatkowska (Subject Editor)
 openscience@royalsociety.org

Associate Editor's comments (Dr Cecilia Mascolo):

The reviewers highlight some improvements which are recommended before this paper can be accepted. In particular there is a recommendation of a better comparison with related work which ought to be followed...

Comments to Author:

Reviewers' Comments to Author:
 Reviewer: 1

Comments to the Author(s)

I enjoyed reading this clearly described and entertaining account of "movements" in art. I thought that the description of your conditional generative adversarial networks was excellent and that you supplied the right level of detail. Clearly, there are many ways in which you could take this work. You might want to speculate about other applications (e.g., weather forecasting or medical prognosis) in your concluding paragraph. You can then point people to the appendix (for another domain of application). I have a few comments that might improve the presentation of your work. Perhaps you could consider the following:

On page 4 (line 32) did you mean "by conditioning on $f_C^{(K+1)}$) as opposed to "sampling from".

Page 4 line 36. Please say "via an automated"

Page 5 (line 9). Replace "and alternative" with "an alternative"

Page 6 (line 33) it might be useful to add something like:

"Note that assuming a fixed (Gaussian) form for the conditional distributions, we are appealing to the same sort of (Laplace) assumption that underwrites variational Bayes. This speaks to the possibility of using approximate Bayesian (i.e. variational) inference to describe (or indeed implement) the current scheme."

On page 8, it might be useful to draw a vertical line on Figure 2 and label the left as the "past" and the right as the "future".

On page 11 (line 19). I was expecting to see a characterisation of the trajectory through the C space. This would be simple to do by plotting the means of each "movement" in the principal eigenvectors of C. This would illustrate the linear progression over successive historical "movements". Perhaps you could think about adding a small figure or inset along these lines. In

other words, supplement or replace Figure 4 using the eigenvectors of the latent dimensions and the centroid of the K mixtures.

Throughout the paper, it would be useful to put “movement” in inverted commas. This is because when you talk about trajectories and artistic movements, it is easy to misread this as an artist “moving” her paintbrush. When you first mention movements, emphasise that these mean “periods” or artistic styles.

I hope that these comments help should any revision be required.

Reviewer: 2

Comments to the Author(s)

I believe this a worthwhile contribution to neural based modelling of arts movements, and a nicely presented article which I'd recommend to be published in the journal should following issues be addressed.

A major missing piece is the lack of presentation of results from any of the related works. I think artificially generated images from collaborative variational autoencoder given the required auxiliary input vector can be compared to the results of your method.

Between-movement and within-movement variation in the experiment presented in Fig.3 can be also quantified statistically (e.g. using entropy related metrics), and not just shown visually. I think Table 1 should be compressed, and more details on the used neural network architecture should be added (as well as presentation and argumentation for simplifications on VGG16 network). Moreover, there are missing references at few points, e.g. line 27 "Unlike previous work in machine learning...".

Finally, it worths exploring whether your learning architecture can be directly extended to model arts movement in the reverse direction, namely towards the past, when time ordering of the input is inverted.

Author's Response to Decision Letter for (RSOS-191569.R0)

See Appendix A.

RSOS-191569.R1 (Revision)

Review form: Reviewer 1 (Karl Friston)

Is the manuscript scientifically sound in its present form?

Yes

Are the interpretations and conclusions justified by the results?

Yes

Is the language acceptable?

Yes

Do you have any ethical concerns with this paper?

No

Have you any concerns about statistical analyses in this paper?

No

Recommendation?

Accept as is

Comments to the Author(s)

Many thanks for attending to my previous suggestions and congratulations on a nice piece of work.

Review form: Reviewer 2

Is the manuscript scientifically sound in its present form?

Yes

Are the interpretations and conclusions justified by the results?

Yes

Is the language acceptable?

Yes

Do you have any ethical concerns with this paper?

No

Have you any concerns about statistical analyses in this paper?

No

Recommendation?

Accept as is

Comments to the Author(s)

I'm happy with the revised version of the article. My comments have been sufficiently addressed and I recommend this work for publication. On a minor correction, there's no correspondence to "Figure (e)" mentioned in the end of page 7.

Decision letter (RSOS-191569.R1)

16-Mar-2020

Dear Mr Lisi,

It is a pleasure to accept your manuscript entitled "Modeling and Forecasting Art Movements with CGANs" in its current form for publication in Royal Society Open Science. The comments of the reviewer(s) who reviewed your manuscript are included at the foot of this letter.

Please ensure that you send to the editorial office an editable version of your accepted manuscript, and individual files for each figure and table included in your manuscript. You can send these in a zip folder if more convenient. Failure to provide these files may delay the

processing of your proof. You may disregard this request if you have already provided these files to the editorial office.

on behalf of Dr Cecilia Mascolo (Associate Editor) and Marta Kwiatkowska (Subject Editor)
openscience@royalsociety.org

Associate Editor Comments to Author (Dr Cecilia Mascolo):
Comments to the Author:

The reviewers are now happy with the changes. Only one comment left: there's no correspondence to "Figure (e)" mentioned in the end of page 7. I trust that you will take care of this before the camera ready version. Congratulations.

Reviewer comments to Author:
Reviewer: 1

Comments to the Author(s)
Many thanks for attending to my previous suggestions and congratulations on a nice piece of work.

Reviewer: 2

Comments to the Author(s)
I'm happy with the revised version of the article. My comments have been sufficiently addressed and I recommend this work for publication. On a minor correction, there's no correspondence to "Figure (e)" mentioned in the end of page 7.

Appendix A

Response to Reviewers

We would like to thank the associate editor and the reviewers for their helpful and constructive comments. Below are point-by-point replies to the comments.

Associate Editor

- *The reviewers highlight some improvements which are recommended before this paper can be accepted. In particular there is a recommendation of a better comparison with related work which ought to be followed...*

- In the following, we explain how we have addressed all the comments from reviewers. We provided new results of additional experiments and more discussion on the different aspects of our proposed method, in comparison to existing conditional generative approaches. A copy of the manuscript that shows modifications in red colour is also included in this submission.

Reviewer 1

- *On page 4 (line 32) did you mean "by conditioning on $f_C^{(K+1)}$ " as opposed to "sampling from".*

This has been updated with more details about what random variable is conditioning on what, namely $X|C$ where C comes from $f_C^{(K+1)}$.

- *Page 4 line 36. Please say "via an automated"*

This has been corrected to say "via an autoencoder".

- *Page 5 (line 9). Replace "and alternative" with "an alternative"*

This has been corrected.

- *Page 6 (line 33) it might be useful to add something like: "Note that assuming a fixed (Gaussian) form for the conditional distributions, we are appealing to the same sort of (Laplace) assumption that underwrites variational Bayes. This speaks to the possibility of using approximate Bayesian (i.e. variational) inference to describe (or indeed implement) the current scheme."*

This has been added to the text in the mentioned location.

- *On page 8, it might be useful to draw a vertical line on Figure 2 and label the left as the "past" and the right as the "future".*

The image has been updated with labels for past and future.

- *On page 11 (line 19). I was expecting to see a characterisation of the trajectory through the C space. This would be simple to do by plotting the means of each "movement" in the principal eigenvectors of C. This would illustrate the linear progression over successive historical "movements". Perhaps you could think about adding a small figure or inset along these lines. In other words, supplement or replace Figure 4 using the eigenvectors of the latent dimensions and the centroid of the K mixtures.*

Figure 4 has been supplemented with an image representing the succession of means of artistic movements in two principal components of the latent space.

- Throughout the paper, it would be useful to put “movement” in inverted commas. This is because when you talk about trajectories and artistic movements, it is easy to misread this as an artist “moving” her paintbrush. When you first mention movements, emphasise that these mean “periods” or artistic styles.

A footnote has been added after the first mention of "movement" to clarify its meaning.

Reviewer 2

- A major missing piece is the lack of presentation of results from any of the related works. I think artificially generated images from collaborative variational autoencoder given the required auxiliary input vector can be compared to the results of your method.

We have expanded our discussion of collaborative variational autoencoders in the last paragraph of the Introduction to clarify why we believe the two methods are not directly comparable. However, we have now included a comparison of our method against a basic method (Appendix B), which will provide the reader with a sense of performance of our algorithm.

- Between-movement and within-movement variation in the experiment presented in Fig.3 can be also quantified statistically (e.g. using entropy related metrics), and not just shown visually.

A matrix of within-cluster variance and between-cluster variance has been added as part of Figure 5, quantifying statistically what is observed in Figure 3.

- I think Table 1 should be compressed, and more details on the used neural network architecture should be added (as well as presentation and argumentation for simplifications on VGG16 network).

The table has been compressed and details on network architectures have been added in the appendix.

- Moreover, there are missing references at few points, e.g. line 27 “Unlike previous work in machine learning...”.

We have added more references.

- Finally, it worths exploring whether your learning architecture can be directly extended to model arts movement in the reverse direction, namely towards the past, when time ordering of the input is inverted.

A discussion of this idea has been added in the text in Section 4.